# Executable Analytic Concepts as the Missing Link Between VLM Insight and Precise Manipulation

## Abstract

Enabling robots to perform precise and generalized manipulation in unstructured environments remains a fundamental challenge in embodied AI. While Vision-Language Models (VLMs) have demonstrated remarkable capabilities in semantic reasoning and task planning, a significant gap persists between their high-level understanding and the precise physical execution required for real-world manipulation. To bridge this "semantic-to-physical" gap, we introduce GRACE, a novel framework that grounds VLM-based reasoning through executable analytic concepts (EAC)—mathematically defined blueprints that encode object affordances, geometric constraints, and semantics of manipulation. Our approach integrates a structured policy scaffolding pipeline that turn natural language instructions and visual information into an instantiated EAC, from which we derive grasp poses, force directions and plan physically feasible motion trajectory for robot execution. GRACE thus provides a unified and interpretable interface between high-level instruction understanding and low-level robot control, effectively enabling precise and generalizable manipulation through semantic-physical grounding. Extensive experiments demonstrate that GRACE achieves strong zero-shot generalization across a variety of articulated objects in both simulated and real-world environments, without requiring task-specific training.

## 1 Introduction

Developing general robotic manipulation systems that can operate effectively in complex, dynamic, and unstructured real-world environments remains a longstanding challenge (Xu et al., 2024). Recent advances in large-scale pretraining have enabled Large Language Models (LLMs) (Naveed et al., 2025; Achiam et al., 2023), including multimodal Vision-Language Models (VLMs) (Zhang et al., 2024; Hurst et al., 2024), to acquire rich world knowledge, demonstrating considerable potential in robotic manipulation tasks. These models are capable of processing complex semantic information and facilitating robust reasoning and planning across diverse scenarios, substantially reducing the dependence on large quantities of high-quality action demonstration data.

Existing VLM-based methods for robotic manipulation have achieved promising results in several areas: task planning (Ahn et al., 2022; Driess et al., 2023), where VLMs interpret natural language instructions and produce high-level action sequences; error detection and recovery (Duan et al., 2024a), where they identify execution failures or environmental anomalies and trigger replanning; and fine-grained action generation (Huang et al., 2025; 2023), where visual representations are extracted and used by VLMs to infer constraints, which are then solved to produce executable robot motions. Another popular approach integrates VLMs with Vision-Language-Action (VLA) models to form a hierarchical architecture: the high-level layer provides semantic reasoning through the VLM, while the low-level layer handles motion planning and execution via the VLA (Ma et al., 2024; Shi et al., 2025).

Despite these advances, VLMs primarily operate within the domain of internet-scale text and 2D images, where they demonstrate strengths in dialogue and static image understanding. However, a significant gap persists between these capabilities and the physical demands of real-world robotic tasks, which is required by precise manipulation within 3D environments. Fine-tuning them into

VLAs is a optional path, yet it is hindered by the high cost of data collection and the risk of creating agent-specific models that lack generalization. Consequently, VLMs struggle to adapt effectively to dynamic settings and complex physical interactions during embodied task execution.

This limitation underscores a fundamental challenge in merging VLMs with robotics: while VLMs reason at a semantic level—interpreting goals and inferring action sequences—robot control operates at the physical level, dealing with forces, velocities, and positions. Bridging this "semantic-to-physical" gap is nontrivial. On one hand, directly embedding LLM-derived knowledge as input features to control policies is often inefficient, as the policy must re-learn physical principles from scratch (Majumdar et al., 2023; Sun et al., 2025). On the other hand, VLMs struggle with the precise numerical reasoning required to express commonsense knowledge in a physically accurate manner, which is essential for tasks demanding high precision (Ahn et al., 2021).

To bridge the semantic knowledge inferred by VLMs and the physical realm in which robots operate, we leverage the notion of analytic concepts (Sun et al., 2024). An analytic concept is a procedural definition, expressed in mathematical terms, that captures the generalized physical commonality of an object or task. When a VLM receives a task prompt and the scene information, we also supply it with a library of concepts. Because the concepts are expressed in precise yet human-readable mathematics, the VLM can weave them naturally into its commonsense chain of thought: it selects the concept that matches the visual evidence, instantiates its free parameters, and determines the semantics of manipualtion. The result is an Executable Analytic Concept (EAC): a blueprint containing grasp poses, force directions, and motion constraints expressed directly in robot coordinates. Within this analytic-concept paradigm the VLM no longer stops at naming objects or describing goals; it assembles a structured, physics-grounded plan whose parameters feed straight into a motion planner, thereby closing the gap between high-level semantics and low-level control.

By mediating between semantic reasoning and physical execution through analytic concepts, our approach leverages the robust commonsense capabilities of LLMs while enabling generalized, interpretable, and precise manipulation of articulated objects. We propose **GRACE** (From VLM-based Grounding to Robotic manipulation through Analytic Concept Execution) with the following contributions:

- We introduce a novel plug-and-play framework that elicits the inherent robotic control potential of VLMs by structured, physics-aware object representations. The framework provides a unified interface that bridges high-level instructions and low-level executable actions for long-horizon manipulation.

- We develop a policy scaffolding pipeline that incorporates analytic concept to translate object-centric semantic knowledge into physically meaningful blueprint, thereby building executable guidance for robot control policies. The executive analytic concepts bridge the gap between VLM's commonsense reasoning and precise physical cognition.

- We demonstrate our approach's outstanding performance in a wide range of manipulation tasks, showcasing the remarkable zero-shot generalization capability in both simulated and real-world environments. We also highlight the compatibility of our EAC-based approach with VLA architecture.

## 2 RELATED WORK

**Structural Representations for Manipulation.** The structural representation chosen for a manipulation system dictates how its modules interact and, consequently, shapes the system's assumptions, efficiency, and overall capability. Traditional approaches rely on rigid-body models: once an object's geometry and dynamics are fully specified, well-understood rigid-body motions can be executed in free space and long-range dependencies are handled efficiently (Migimatsu & Bohg, 2020; Dantam et al., 2018). Yet this strategy presupposes that accurate geometry and physical parameters of the environment are available a priori—a requirement rarely met outside carefully curated setups. To relax this constraint, recent research has explored data-driven alternatives, including learned object-centric embeddings (Hsu et al., 2023; Cheng et al., 2023; Yuan et al., 2022), particle-based modeling (Bauer et al., 2024; Abou-Chakra et al., 2024), and keypoint or descriptors (Simeonov et al., 2022; Manuelli et al., 2019; Huang et al., 2024b). Although promising, these approaches often suffer from instabil-

ity, manual annotation, or a reliance on hand-crafted geometric priors, limiting their reliability and breadth of application.

**Vision-Language Models for Robotics.** Our work builds upon recent advances in Vision-Language Models (VLMs) for robotic control, which demonstrate remarkable capabilities in scene understanding and high-level commonsense reasoning. Existing approaches can be broadly categorized into several paradigms (Shao et al., 2025). Some studies integrate environmental perception—including visual, linguistic, and robot state information—along with action generation into a unified Visual-Language-Action (VLA) model (O'Neill et al., 2024; Zitkovich et al., 2023; Deng et al., 2025). Alternatively, dual-system architectures employ a VLM backbone for scene interpretation and a separate action expert for policy generation, communicating through latent representation exchanges. Despite their promise, these methods often require large-scale data collection and face challenges in generalizing beyond training distributions. Other efforts seek to leverage visual foundation models to extract operational primitives, which then serve as visual or linguistic prompts to VLMs for task-level reasoning (Duan et al., 2024b; Huang et al., 2024a; Pan et al., 2025). These systems typically rely on traditional motion planners for low-level control. However, such approaches are limited by the loss of geometric detail when compressing 3D physical interactions into 2D images or 1D textual descriptions, as well as by the inherent hallucination problems of VLMs. These limitations often compromise the accuracy and executability of high-level plans generated by VLMs.

Addressing these challenges, we introduce analytic concepts as a core component that scaffolds the VLM's reasoning process, enabling it to progressively derive physical knowledge of objects from fine-grained 3D geometric information and produce executable and accurate manipulation plans.

## 3 ANALYTIC CONCEPTS

The analytic concepts take inspiration from the advancements of researches on human cognition and brain science, where it is discovered that we humans learn about the physical world by perceiving geometry patterns from objects and inducing them along with related knowledge as commonsense for future reference. Based on such findings, a novel knowledge annotation paradigm for object understanding tasks is established by explicitly modeling such abstract commonsense information as concepts for regular geometry patterns and reversing the induction process (Sun et al., 2024). Specifically, by generalizing the concepts towards certain objects, various knowledge associated with the concepts can be automatically propagated to all these objects.

In engineering and architecture, a blueprint is a detailed plan that defines the structure of an object through specifications and guides its fabrication and assembly. We introduce analytic concepts to play an analogous role for robots: they are procedural, mathematics-based definitions that capture the shared physical essence of an object or its sub-components, turning abstract knowledge into an *executable blueprint* for manipulation. At their foundation, analytic concepts include a "factory" of geometric concept assets (Fig. 1a). Each asset code provides a set of free parameters to represent diverse variations, a canonical structural definition, and affordance annotations as concise descriptors of how the object can be grasped or acted upon. Besides, a function is also provided to render instances of the assets in 3D space. These assets are the atomic building blocks from which every executable blueprint is assembled with building structural blueprint and manipulation blueprint.

The analytic structural blueprint is a series of mathematical procedures revealing the essential commonality of the spatial structure, including spatial layout and structural relationships, shared by all instances of the concept, as shown in Fig. 1b. Further, there are variable parameters in the procedures to represent the variations among different physical instances. That is, a physical instance of this concept can be created with specific parameters, and in turn, a target in the physical world can be also resolved into parameters of a concept.

Effective interaction requires more than geometric fidelity; it demands knowledge of functional properties such as affordances and force dynamics. To this end, we can ground manipulation blueprint (Fig. 1c) that meet the functional properties of the concept and force directions that would cause effective movement. Similarly to the analytic structural blueprint, the analytic manipulation blueprint is also formulated by mathematical procedures with variable parameters. It may incorporate multiple interaction strategies, each accompanied by a precise natural-language synopsis to facilitate high-level reasoning by language models.

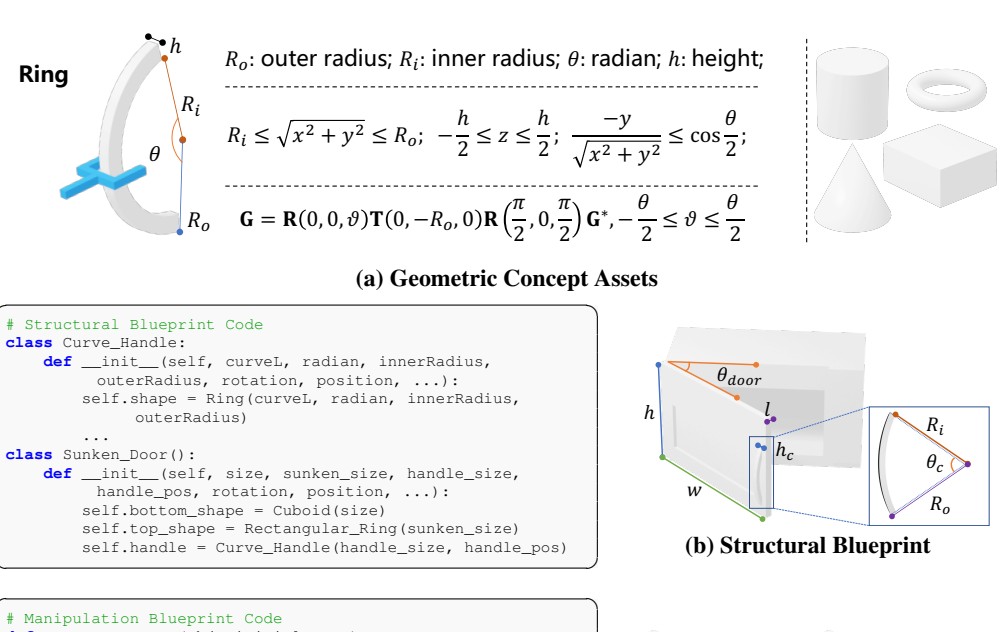

**(a) Geometric Concept Assets**

```
# Structural Blueprint Code
class Curve_Handle:
    def __init__(self, curveL, radian, innerRadius,
        outerRadius, rotation, position, ...):
        self.shape = Ring(curveL, radian, innerRadius,
            outerRadius)
        ...
class Sunken_Door():
    def __init__(self, size, sunken_size, handle_size,
        handle_pos, rotation, position, ...):
        self.bottom_shape = Cuboid(size)
        self.top_shape = Rectangular_Ring(sunken_size)
        self.handle = Curve_Handle(handle_size, handle_pos)
```

**(b) Structural Blueprint**

```
# Manipulation Blueprint Code
def get_grasp_pose(obj, initial_pose):
    if (isinstance(obj, Curve_Handle)):
        local_pose = translate_world2local(pose, init_pose)
        grasp_pose = obj.apply_pose(local_pose)
    if (isinstance(obj, Sunken_Door)):
        ...
    return translate_local2world(grasp_pose)
def push(obj):
    force_direction = get_direction(obj.axis...)
def pull(obj):
    force_direction = get_direction(obj.axis...)
```

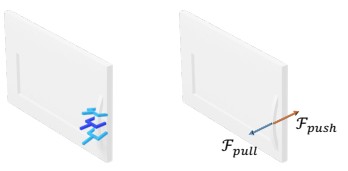

**(c) Manipulation Blueprint**

Figure 1: Example implementation of executable analytic concepts. (a) Geometric Concept Assets. Each asset exposes its free parameters (top), canonical structure (mid), and partial affordance cues (bottom). (b) Structural Blueprint: higher-level objects are procedurally composed by wiring multiple geometric assets together, forming a parametric graph that captures their spatial layout and structural relationships. (c) Manipulation Blueprint: parameterised routines compute grasp poses and force directions that exploit the affordances encoded in the underlying structure.

## 4 METHODOLOGY

**Problem Formulation.** This paper addresses the challenge of enabling a robotic system to perform manipulation tasks based on high-level language instructions. Our system is given a visual observation $O_t$ of the environment and a natural language instruction $l$ describing the desired task. The core difficulty lies in bridging the gap between high-level human commands and low-level physical actions due to the complexity of the object operated. The language instruction $l$ can be both arbitrarily long-horizon and under-specified, requiring the system to possess advanced commonsense reasoning to infer user intent and contextual details. To successfully complete the task with a parallel gripper, the robot must not only understand the object and task description but also manage the complex physics of contact-rich interactions. This necessitates an intelligent system capable of generating precise affordances and robust grasp strategies.

**Overview** As illustrated in Figure 2, the proposed GRACE framework orchestrates a pipeline built around a Vision-Language Model (VLM) that transforms a natural language instruction and an RGB-D image into a successful robot action. The process begins with (I) Task Parsing, where the VLM parses and comprehends the user command (e.g., "Open the upper handle.") within the visual context of the observed scene. The core contribution of our work lies in (II) Policy Scaffolding, a sophisticated VLM-driven process that constructs an Executable Analytic Concept (EAC). This is accomplished through a structured sequence: first segmenting the target point cloud, and then grounding both structural and manipulation blueprint. Finally, the VLM performs reasoning over this rich, structured EAC to generate precise motion parameters, which are subsequently passed to the mo-

Figure 2: An overview of the proposed method GRACE. (I) **Task Parsing**: A Vision–Language Model (VLM) parses the natural-language instruction based on the current RGB image. (II) **Policy Scaffolding**: The process includes: 1. segmenting the target object from images and back-projecting it to a partial point cloud; 2. parsing the analytic concept and estimating geometric parameters to instantiate the structural blueprint; 3. constructing the manipulation blueprint to produce feasible grasp poses and force directions; 4. generating a joint-space trajectory via a motion-planning module using the blueprints. (III) **Robot Execution**: The trajectory is executed to complete the task.

tion planner for (III) Robot Execution. The EAC acts as the essential missing link that grounds the VLM's abstract "insight" into a physically precise and executable format.

## 4.1 SPATIAL-AWARE TASK PARSING

**Object Parsing.** The Object Parsing step serves as the foundational stage for perception and language grounding. Its objective is to interpret the natural language instruction $l$ within the context of the RGB-D scene images, producing a structured set of task-relevant object entities along with their critical spatial information. This process distills the "what" and "where" from the command, delivering a clean symbolic input for downstream task reasoning and planning.

We implement the parsing through a structured chain-of-thought (CoT) reasoning process with two core steps: (i) The VLM first performs a coarse-to-fine analysis to identify primary objects, extracting noun phrases and their synonymous references grounded in the visual scene layout. (ii) The VLM then assesses object states—particularly for articulated objects—and identifies binary spatial relationships between entities. The final output is a structured graph $\mathcal{G} = (\mathcal{V}, \mathcal{E})$, where $\mathcal{V}$ denotes the set of object nodes—each represented as a structured dictionary containing id, name, and state—and $\mathcal{E}$ constitutes a set of directed spatial relationships between objects, each expressed as a triple $e_{ij} = (v_i, r, v_j)$. This object-centric symbolic graph provides a semantically rich and structurally explicit representation for subsequent reasoning stages.

**Task Decomposition.** For complex, long-horizon tasks, our approach first decomposes the primary task into a series of stages, each defined by object interaction primitives with associated spatial constraints. Subsequently, a VLM, leveraging object parsing information, is used to decompose the main task instruction l into a series of discrete sub-tasks, represented as $l_i$, along with a corresponding verification condition $c_i$, for $i \in \{1, \ldots, n\}$. This transforms the instruction l into a sequence of specific sub-tasks and conditions: $\{(l_1, c_1), (l_2, c_2), \ldots, (l_n, c_n)\}$. For instance, the high-level task "open the microwave door" could be decomposed into sub-tasks like "grasp the door handle" and "pull open the door," with verification conditions such as "is the handle grasped?" and "is the door opened?". Each sub-task then undergoes an execution loop, as depicted in Fig. 2. After the initial execution attempt, the task reasoning program is replaced with a corresponding condition verification program to ensure the successful completion of that sub-task. This structured approach allows for the precise definition of task requirements and facilitates the execution of complex manipulation tasks. See Appendix D for prompts.

## 4.2 POLICY SCAFFOLDING

Policy scaffolding as core first determines the target object or part that needs to be analyzed, and then builds the structural and manipulation blueprint in turn to obtain the executable analysis concept.

### 4.2.1 TARGET IDENTIFICATION

In the object parsing step, we obtain a structured object graph $\mathcal{G} = (\mathcal{V}, \mathcal{E})$. Using the names from $\mathcal{V}$ as object category prompts, we leverage Visual Foundation Models (VFMs) to perform open-vocabulary instance segmentation. Specifically, GroundingDINO (Liu et al., 2024) localizes referred objects, and the Segment Anything Model (SAM) (Kirillov et al., 2023) generates fine-grained 2D masks $\mathcal{M} = \{M_i \mid i = 1, 2, \ldots, m\}$ for all foreground objects relevant to the task. Each 2D mask $M_i$ is then back-projected into 3D using the corresponding depth image, producing a set of object-centric 3D point clouds $\mathcal{P} = \{P_i \mid i = 1, 2, \ldots, m\}$. These point clouds are associated with the semantic nodes $v_i \in \mathcal{V}$, effectively grounding the symbolic elements of $\mathcal{G}$ into geometrically precise representations.

### 4.2.2 STRUCTURAL BLUEPRINTING

With the obtained target part's point cloud $\mathcal{P}$, we proceed to ground its geometric structure in a formalized representation. We do so by querying a pre-defined library of analytic concepts, which are parameter-driven models that capture common structural archetypes (e.g., primitive geometries, typical handle designs), each paired with a short natural-language synopsis. For example, in the Fig. 1(b), take the concept of ring, which frequently appears in the design of handles, by discovering the ring concept on a handle as an analytic description, we can identify its size (e.g., inner radius and outer radius) and pose, as well as the detailed parameters for the orientation of its hinge. The grounding procedure unfolds in two successive stages. First, we prune the concept library according to the part category detected in the previous step, and prompt the VLM with the synopses of the remaining candidates, asking: "Find the part to interact within <target object> the in order to complete the task <sub-task>, and determine the <concept> of the part." This query lets the VLM map its high-level semantic perception onto a node in our geometric knowledge graph, thereby fixing the symbolic layout of the structural blueprint.

Next, we must turn that symbolic layout into an executable program by instantiating every node with concrete parameters, estimated directly from the point cloud $\mathcal{P}$. These parameters are of two types:

- **Structural parameters** encode the concept's intrinsic geometry of the analytic concept (e.g., the size $l, w, h$ of a sunken door). To estimate them, we encode the point cloud $\mathcal{P}$ into a deep feature vector using an encoder. This feature vector is then fed into multiple specialized MLP heads, each regressing a specific structural parameter.

- **6-DoF pose parameters** locate the concept's global position and orientation. These are recovered analytically by combining the object's known simulation pose with the newly estimated structural variables.

### 4.2.3 MANIPULATION BLUEPRINTING

The structural blueprint tells us *what* the target part is; the manipulation blueprint specifies *how* to interact with it. Affordances of geometric ontologies are encoded as analytic manipulation knowledge for grasp poses, pushing contacts, and similar actions, while kinematic ontologies additionally provide force directions that produce motion. All of this knowledge is expressed by mathematical formulas with tunable parameters and offers critical guidance for downstream control.

We begin by presenting the VLM with the natural-language synopses of every candidate manipulation function—e.g., "pull-type grasp on curve handle," "push at door edge." The VLM chooses the module that best fulfils the high-level goal ("open the microwave door") and returns its analytic form. In this way, the model's semantic understanding is mapped directly onto executable actions.

Each selected function defines a category of grasp poses belonging to the same pattern. An exact grasp pose $\mathbf{G}$ is physically grounded by estimating the parameters of such analytic knowledge. Different from the structural parameters which are unique for a specific part, grasp-pose parameters have multiple valid solutions. For optimal door operation, grippers typically interact with the handle

within its designed graspable range. However, under certain circumstances, the door edge itself also presents functional affordances that enable operation. With the parameters, a physically grounded grasp pose $\mathbf{G}$ can be calculated according to the analytic manipulation knowledge and initial grasp pose $\mathbf{G}^*$. For example, the equation

$$\mathbf{G} = \mathbf{R}(0, 0, \vartheta)\mathbf{T}(0, -R_o, 0)\mathbf{R}(\frac{\pi}{2}, 0, \frac{\pi}{2})\mathbf{G}^*, -\frac{\theta_c}{2} \leq \vartheta \leq \frac{\theta_c}{2}$$

indicates a function that transforms the initial gripper pose to a grasp pose for the curve handle shown in Fig. 1(b). Once $\mathbf{G}$ is fixed, the force-direction formula—conditioned by the verb or manipulation type chosen by the VLM (e.g., *pull* vs. *push*)—is invoked to produce the vector $\mathcal{F}$, ensuring that the applied force is semantically aligned with the selected action and correctly oriented on the target part. Both $\mathbf{G}$ and $\mathcal{F}$ are exported as lightweight Python functions and fed to the physically-grounded evaluator, closing the loop from language to low-level control.

### 4.3 Low-Level Motion Execution

**Blueprint Execution.** The instantiated structural and manipulation blueprints jointly output two quantities in the *local* frame of the target part: a grasp pose $\mathbf{G}_{\text{local}} = (\boldsymbol{t}_{\text{local}}, \boldsymbol{r}_{\text{local}})$, and a force direction $\mathcal{F}_{\text{local}}$. Running the blueprint therefore reduces to transforming these local descriptors into the world frame and then feeding them to a standard motion–planning stack.

**Transformation to World Coordinates.** Let $\mathbf{M} \in \mathbb{R}^{4 \times 4}$ denote the homogeneous transform of the target part with respect to the world frame, obtained from perception or simulation. For every point–set or inequality description $F$ in the blueprint we apply $F\big((x, y, z, 1)^\top\big) \leq 0 \implies F\big(\mathbf{M}^{-1}(x, y, z, 1)^\top\big) \leq 0$, thereby re-expressing all structural constraints globally. The grasp pose is mapped by $\mathbf{G}_{\text{world}} = \mathbf{M}\mathbf{G}_{\text{local}}$. For rotationally symmetric geometries we additionally enforce a minimal-rotation constraint on $\boldsymbol{r}_{\text{local}}$ to obtain a unique orientation. The force vector is transformed analogously: $\mathcal{F}_{\text{world}} = \mathbf{R}\,\mathcal{F}_{\text{local}}$, where $\mathbf{R}$ is the rotational part of $\mathbf{M}$.

**Motion Planning and Execution.** The world-frame grasp pose $\mathbf{G}_{\text{world}}$ and force vector $\mathcal{F}_{\text{world}}$ are forwarded to a low-level planner. The planner first synthesises a collision-free approach path, then a compliant trajectory to realise the grasp, and finally an interaction phase that applies a wrench aligned with $\mathcal{F}_{\text{world}}$. The resulting joint-space command sequence is streamed to the robot controller, closing the pipeline from high-level language to physical motion.

## 5 Experiments

To comprehensively evaluate the effectiveness and generalization capability of our proposed GRACE framework, we conduct extensive experiments in both simulated and real-world environments. This section is organized as follows: We begin with a zero-shot manipulation evaluation in simulation in Section 5.1. In order to verify the structural understanding of articulated objects by the process of policy scaffolding, additional interactive experiments are carried out in Section 5.2. We also carry out the object manipulation experiments with physical robots in real-world environments to provide a more comprehensive and stronger evaluation in Section 5.3. We provide implementation details of GRACE in Appendix A.

### 5.1 Manipulation Evaluation in Simulation

We select SimplerEnv (Li et al., 2024c) as our simulation platform due to its open-source nature and its focus on real-world robotic manipulation. It offers a standardized benchmark suite that emphasizes reproducible results and maintains close alignment with physical hardware constraints and realistic task conditions. We conduct quantitative evaluations of GRACE's zero-shot execution performance on Google Robot tasks & Widow-X tasks and compare it to baselines including Octo (Ghosh et al., 2024), OpenVLA (Kim et al., 2024) and more concurrent works (Qi et al., 2025; Qu et al., 2025; Li et al., 2024b).

On the four Widow-X tasks (Table 1), GRACE powered by GPT-4o achieves an average success rate of 86.1%, clearly outperforming the strongest published baseline, SoFar (58.3%). Although it is not the best on every single task, GRACE never performs poorly, maintaining consistently high scores

Table 1: **SimplerEnv simulation evaluation results for the WindowX Robot task.** We report both the final success rate ("Success") along with partial success (e.g., "Grasp Spoon"). "FT" denotes performance of the fine-tuned models.

| Model | Put Spoon on Towel | | Put Carrot on Plate | | Stack Green Block on Yellow | | Put Eggplant in Basket | | Avg |
|---|---|---|---|---|---|---|---|---|---|
| | Grasp Spoon | Success | Grasp Carrot | Success | Grasp Block | Success | Grasp Eggplant | Success | |
| RT-1-X | 16.7% | 0.0% | 20.8% | 4.2% | 8.3% | 0.0% | 0.0% | 0.0% | 1.1% |
| Octo-small | 77.8% | 47.2% | 27.8% | 9.7% | 40.3% | 4.2% | 87.5% | 56.9% | 30.0% |
| OpenVLA | 4.1% | 0.0% | 33.3% | 0.0% | 12.5% | 0.0% | 8.3% | 4.1% | 1.0% |
| RoboVLM | 37.5% | 20.8% | 33.3% | 25.0% | 8.3% | 8.3% | 0.0% | 0.0% | 13.5% |
| RoboVLM (FT) | 54.2% | 29.2% | 25.0% | 25.0% | 45.8% | 12.5% | 58.3% | 58.3% | 31.1% |
| SpatialVLA | 25.0% | 20.8% | 41.7% | 20.8% | 58.3% | 25.0% | 79.2% | 70.8% | 34.4% |
| SpatialVLA (FT) | 20.8% | 16.7% | 29.2% | 25.0% | 62.5% | 29.2% | **100.0%** | **100.0%** | 42.7% |
| SoFar | 62.5% | 58.3% | 75.0% | 66.7% | **91.7%** | 70.8% | 66.7% | 37.5% | 58.3% |
| SpatialVLA-EAC | **91.7%** | **87.5%** | 79.2% | 62.5% | 75.0% | 50.0% | 79.2% | 79.2% | 69.8% |
| GRACE(Qwen2.5-VL) | 83.3% | 83.3% | **79.2%** | **79.2%** | 87.5% | 83.3% | 91.7% | 91.7% | 84.4% |
| GRACE(GPT-4o) | 83.3% | 83.3% | **79.2%** | **79.2%** | 87.5% | **87.5%** | 95.8% | 95.8% | **86.1%** |

Table 2: **SimplerEnv simulation evaluation results for the Google Robot setup.** We present success rates for the "Variant Aggregation" and "Visual Matching" approaches. "FT" denotes performance of the fine-tuned models.

| Model | Variant Aggregation | | | Visual Matching | | | Avg |
|---|---|---|---|---|---|---|---|
| | Pick Coke Can | Move Near | Open/Close Drawer | Pick Coke Can | Move Near | Open/Close Drawer | |
| RT-1-X | 49.0% | 32.3% | 29.4% | 56.7% | 31.7% | 59.7% | 43.1% |
| Octo-Base | 0.6% | 3.1% | 1.1% | 17.0% | 4.2% | 22.7% | 8.11% |
| OpenVLA | 54.5% | 47.7% | 17.7% | 16.3% | 46.2% | 35.6% | 36.3% |
| RoboVLM | 68.3% | 56.0% | 8.5% | 72.7% | 66.3% | 26.8% | 49.8% |
| RoboVLM(FT) | 75.6% | 60.0% | 10.6% | 77.3% | 61.7% | 43.5% | 54.8% |
| SpatialVLA | 89.5% | 71.7% | 36.2% | 81.0% | 69.6% | 59.3% | 67.9% |
| SpatialVLA(FT) | 88.0% | 72.7% | 41.8% | 86.0% | 77.9% | 57.4% | 70.6% |
| SoFar | 90.7% | 74.0% | 29.7% | **92.3%** | **91.7%** | 40.3% | 69.6% |
| SpatialVLA-EAC | 88.9% | 77.9% | 83.3% | 86.1% | 79.2% | 85.4% | 83.4% |
| GRACE(Qwen2.5-VL) | 90.3% | 87.5% | 88.9% | 91.7% | 88.9% | 84.7% | 88.7% |
| GRACE(GPT-4o) | **91.7%** | **87.5%** | **90.3%** | 90.3% | **91.7%** | **88.9%** | **90.1%** |

across the entire suite. The pattern repeats on the Google-robot tasks (Table 2): GRACE(GPT-4o) attains 89.8% mean success, exceeding the best prior result by almost 30 pp. Notably, on the articulated Open/Close Drawer task the jump is the largest, rising from 29.7% (SoFar) and 36.2% (SpatialVLA) to 90.3% with GRACE for "Variant Aggregation", highlighting the advantage of EACs when precise kinematic reasoning is required.

To isolate the contribution of analytic concepts, we retrofit SpatialVLA by replacing its native, end-to-end action output with EAC-guided motion planning when the gripper approaches the target; this variant is denoted *SpatialVLA-EAC*. The simple swap boosts SpatialVLA's average success to 69.8% on Widow-X and to 83.4% on the Google robot, demonstrating that EACs can be used as a plug-and-play module to substantially enhance existing VLA architectures. Finally, GRACE's performance is insensitive to the underlying VLM. The fully open-source Qwen2.5-VL backend trails GPT-4o by only 1–2 pp on both robot families, yet still outperforms every external baseline, confirming that the bulk of the gain comes from the analytic-concept layer rather than the choice of language model.

## 5.2 Manipulation Experiment of Articulated Objects

To focus on articulated objects manipulation, we evaluate the GRACE through the success rate of interaction on the proposed task, i.e., changing an articulated object from its initial state to

a target final state. The success rate can reveal the quality of articulated concept discovery, including ontology discovery and affordance grounding. All experiments are carried out in SAPIEN under the standard Where2Act (Mo et al., 2021) settings (Appendix B for detail). We compare our method against three baselines, i.e., Where2Act, Where2Explore (Ning et al.) and ManipLLM (Li et al., 2024a), each representative of a distinct modelling paradigm for articulated–object manipulation. To isolate the contribution of VLM reasoning, we also report an ablated variant, GRACE-w/o-VLM, in which the concept-selection step is replaced by ground-truth ontology labels.

Table 3 demonstrates that GRACE(GPT-4o) achieves the highest scores across all categories. For instance, it attains 0.65 for "faucet" objects and 0.91 on "cabinet" doors, significantly outperforming ManipLLM, which scores 0.26 and 0.71, respectively. These results decisively surpass both pixel-level affordance methods and the LLM-based ManipLLM. The substantial numerical margins underscore the advantage of integrating VLM-based reasoning with analytically grounded control. Replacing the oracle concept label with GPT-4o's automatic selection reduces performance only slightly—from an average of 0.80 to 0.77, a drop of roughly three percentage points. The small gap indicates that the few remaining failures are due primarily to occasional VLM misclassification rather than limitations of the analytic concepts themselves; once the correct concept is chosen, execution is highly reliable.

Table 3: Comparison of performance on different objects (icons represent object categories).

| Objects | 📦 | 🗄 | 🚪 | 💻 | 📺 | 🔧 |
|---|---|---|---|---|---|---|
| Where2Act | 0.14 | 0.68 | 0.27 | 0.23 | 0.15 | 0.15 |
| UMPNet | 0.44 | 0.54 | 0.28 | 0.54 | 0.28 | 0.25 |
| ManipLLM | 0.65 | 0.71 | 0.77 | 0.43 | 0.65 | 0.26 |
| w/o-VLM | **0.85** | **0.91** | **0.90** | **0.70** | **0.78** | **0.65** |
| (GPT-4o) | 0.84 | 0.85 | 0.88 | **0.70** | 0.72 | 0.60 |

## 5.3 OBJECT MANIPULATION EVALUATION IN REAL-WORLD

We conducted experiments in a real-world tabletop environment using a Realman RM75 robotic arm equipped with a parallel gripper. Detailed visualizations of the environment and additional robot setup specifications are provided in Appendix B. For qualitative analysis, we first visualize the outputs and success rate of our approach for four different objects in Fig. 3, demonstrating the promising zero-shot manipulation capability of EAC for physics-grounded planning. Experimental results indicate that the VLM only needs to identify the target part of an object and construct its EAC representation to enable the robot to successfully complete the task. To further thoroughly assess the generalization ability of GRACE, we designed a long-

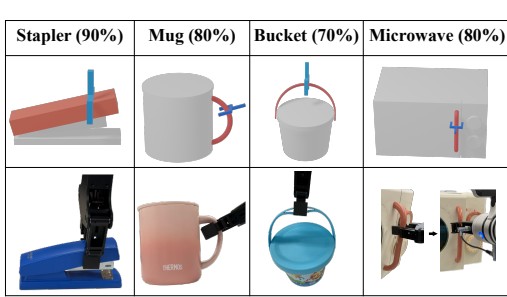

Figure 3: Visualize the results of grasping objects and their corresponding EAC. The red parts in the second column indicate the target part.

horizon manipulation task involving six diverse objects. Preliminary observations suggest that GRACE maintains robust task reasoning capabilities even as task complexity increases. The overall performance in this long-horizon task is presented in the supplementary video.

## 6 CONCLUSION

We have introduced GRACE, a plug-and-play framework that grounds visual observations with a VLM, reasons over Executable Analytic Concepts, and converts the result into precise robot actions. Extensive experiments on simualtion and real world demonstrate marked gains in zero-shot success rates, particularly on kinematically challenging tasks. In future work we plan to extend analytic concepts to multi-fingered hands and to explore on-the-fly concept refinement from real-world interaction data.

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

## A  IMPLEMENTATION DETAILS OF METHOD

**Segmentation.**   We use Grounded-SAM (Ren et al., 2024) consisting of two major components, Grounding-Dino (Liu et al., 2024) and SAM (Kirillov et al., 2023). We keep SAM frozen and fine-tune Grounding-Dino with RGB images with ground-truth bounding boxes of the actionable objects or parts, along with natural language prompt that describes the actionable objects or parts provided by VLM.

**Parameter Estimation.**   The encoder is a Point-Transformer that extracts 128 groups of points with size 32 from the input with 2048 points and has 12 6-headed attention layers. The subsequent MLP has three layers with ReLU activation and outputs the structural parameters. The network is trained with L2 loss between the estimated and ground-truth structural parameters. Throughout the operation of the GRACE framework, the model parameters remain fixed. To construct the training dataset for our models, we first create analytic concept annotations for real-world objects. Specifically, we label the concept parameters of the training objects from PartNet-Mobility. Each object is then imported into the SAPIEN simulator, where a virtual camera captures RGB images and depth maps. Using the object's URDF file together with our analytic annotations, we can automatically generate ground-truth data—including bounding boxes, point clouds and structural parameters for every actionable part. Additionally, we leverage the FoundationPose (Wen et al., 2024) model for 6D object pose estimation.

## B  EXPERIMENTAL SETUP

**Articulated Objects Manipulation Setup**   All evaluations are carried out in the SAPIEN [33] physics simulator. At the start of each manipulation episode, the target object is placed at the scene origin. Its articulated joint is initialized randomly: there is a 50 % chance of starting in the fully closed configuration and a 50 % chance of starting in a random open configuration. An RGB-D camera with known intrinsics is aimed at the scene centre from a point sampled on the upper hemisphere, with azimuth uniformly drawn from $[0°, 360°)$ and elevation from $[30°, 60°]$. Interaction is performed with a two-finger "flying" Franka Panda gripper. We restrict the controller to two primitive actions: pushing and pulling. A flying Franka-Panda gripper serves as the agent, and perception is obtained from a single RGB-D camera placed five units from the object centre.

**Real World Robot Setups**   We detail our hardware setup in Figure 4, which centers on a Realman RM75 Arm. For perception, we integrate a single RGB-D camera (Intel RealSense D435) mounted on the end-effector. The system is powered by a workstation equipped with an Intel Core i9-14900K processor, 64GB of RAM, and an NVIDIA RTX 4090 GPU, ensuring real-time inference and planning.

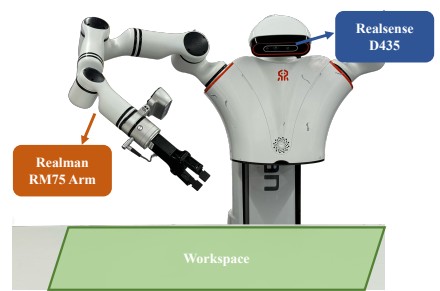

Figure 4: Hardware Configuration.

**Long-horizon Task**   We design a long-horizon task to validate the capabilities of our framework. All the objects being manipulated are not seen by the model. The task instruction is: *tidy up the table and open the microwave.* The overall performance in this long-horizon task is presented in the supplementary video.

## C  SYSTEM ERROR BREAKDOWN

The primary sources of failure in our system are pose estimation and inverse kinematics (IK). Our analysis indicates that employing multi-view images for 3D object reconstruction significantly enhances the success rate of pose estimation. It is also recommended to use high-resolution cameras to further improve estimation accuracy. Although structural parameter estimation introduces some

error, its impact on the overall success rate is relatively minor. In contrast, the VFM-based object grounding module, alongside the VLM-based task parsing and concept construction, demonstrates high stability and contributes negligibly to system failures.

## D PROMPTS FOR TASK PARSING

```
Task_Parsing_PROMPT_TEMPLATE_1 = """
**Role:** You are an expert robotic task planner. Your job is to analyze
    a visual scene image and break down a high-level manipulation command
     into a sequence of low-level, executable actions for a robot arm
    equipped with a gripper.
**Task:** {task}
**Example:** Task: "Pour the water from the blue cup into the red mug."
**Scene Image Context:**
the given image
**Robot Capabilities:**
- The robot has a single arm with a parallel-jaw gripper.
- It can perform primitives: grasp(object_name), lift(height), pour(
    into_object_name), place_on(object_name), release(), push(object_name
    ), pull(object_name).
- It cannot perform actions requiring complex dexterity (e.g., tying
    knots, unscrewing tight lids).
- It must avoid collisions with all objects not involved in the task.

**Output Instructions:**
1.  **Reasoning:** First, reason step-by-step. Identify the key objects
    involved and their properties. The final output is a  structured
    object graph G = (V, E), where V denotes the list of object nodes,
    each represented as a structured dictionary  containing id, name, and
     state, and E constitutes a list of directed spatial relationships
    between objects, each expressed as a triple e = (vi, r, vj).
2.  **Plan:** Based on your reasoning, generate a sequence of action
    commands. The sequence must be logical, safe, and efficient. Each
    action instruction must include a validation condition that can be
    understood, such as verifying the target object is successfully
    grasped.
3.  **Final Output:** Provide **only** a valid JSON array as the final
    output. Do not add any other text. The JSON must follow this schema:
 json
    {{
      "task": "original_task_description",
      "objects_graph_V": "structured object list",
      "objects_graph_E": "structured object spatial relationships list",
      "action_instruction_sequence": [
        {{"id": 1, "action": "action_name", "parameter": "
            target_object_or_value", "success":"validation_condition"}},
         {{"id": 2, "action": "action_name", "parameter": "
            target_object_or_value", "success":"validation_condition"}}
      ]
    }}

**Now, analyze the provided scene image and complete the task.**
"""

Task_Parsing_PROMPT_TEMPLATE_2 = """
**Role:** You are a robotic task completion verifier. Your job is to
    analyze whether a manipulation task has been successfully completed
    by comparing the current scene state with the expected goal state.

**Original Task:** "{Origin_Task_Description}"

**Expected Goal State Description:**
{Validation_Condition}
```

```
**Scene Image Context:**
the given image

**Final Output:**
Provide **only** a valid JSON array as the final output. Do not add any
    other text. The JSON must follow this schema:
json
    {{
  "task_completed": boolean,
  "error_message": string
}}
"""
```

## E   STATEMENT ON LARGE LANGUAGE MODEL USAGE

This paper employed Large Language Models to assist in the writing process. The LLM was used exclusively for the purpose of language polishing, which included:

- Correcting grammatical errors.
- Improving sentence fluency and readability.
- Refining word choice for better academic tone.

The LLM was **not** used for generating original ideas, formulating research hypotheses, conducting data analysis, or interpreting results. All intellectual content and scholarly contributions are solely those of the authors. The authors have thoroughly reviewed, revised, and take complete responsibility for the entire content of this manuscript.

