# OpenReview forum: "Executable Analytic Concepts as the Missing Link Between VLM Insight and Precise Manipulation"
_ICLR.cc/2026/Conference — ICLR 2026 Conference Withdrawn Submission_

### Official Review · Reviewer_jLCA · 2025-10-27

**Soundness:** 3
**Presentation:** 2
**Contribution:** 2
**Rating:** 2
**Confidence:** 4

**Summary:**

This paper proposes a VLM-based framework for robot manipulation of articulated objects, that aims to balance generalization and precision. It proposes the "executable analytic concept" that represents an object's geometry, structure, and manipulation blueprint. Based on this, it leverages the VLM to parse high-level task instructions in natural languages into executable action plans related to the object.

**Strengths:**

- The paper provides a good summary of how VLMs are currently used in robot learning, and points out their struggles in balancing generalization and knowledge of real-world physics/dynamics/motion (Introduction paragraph 2-3). These high-level discussions sound very reasonable and provide a good motivation for the work.
- It proposes a practical framework, with improved performance in several tasks.

**Weaknesses:**

**Writing**

To my feeling, the writing focuses too much on high-level storytelling, but misses too many problem details. I think making these things clear is particularly important, so that there could be a clear boundary for what problems this work can solve and what problems it cannot -- given that generalization is a main claim in this paper.
Concretely:

*Section 3 Analytical Concepts*

I think this section is important as the "analytical concept" is a core term proposed in this paper. However, I feel Section 3 fails to give a clear enough definition of it. The only concrete description of it is probably Figure 1. However, although Figure 1 provides a straightforward explanation of what it looks like, I think it doesn't actually give a clear term definition. What's missing:
	- What are exactly the components and variables in "executable analytic concepts"?
	- What are the problem settings it can model? (And what settings can it not model?) For example: In its geometric component, can it model freeform shapes represented as triangle meshes, or it has to be parametric shapes? In its manipulation component, what actions can be modeled, in addition to pull/push? The force has to be point-based (so that it's mostly for a parallel gripper instead of a dexterous hand)?

*Section 4 Methodology -- Problem Formulation*

Similarly, I feel the problem formulation part is also not clear enough. For example:
	- It mentions that the important problem of "bridging the gap between high-level human commands and low-level physical actions". I understand that the "high-level human commands" mean the language instruction $l$ (which can be both arbitrarily long-horizon and under-specified). But what are the "low-level physical actions"? -- contact maps, gripper poses, etc.? Is it a single point-based force direction, or is it a motion sequence (with the temporal dimension)?
	- It says that the system is given a visual observation $O_t$. The subscript $t$ means time? So the observation is a temporal sequence?

*Section 4 Methodology (4.1-4.3)*

I also have the same feeling when reading the rest of Section 4.

How is the object parsed? I don't see a separate prompt, but just the task-parsing prompt in Appendix D. Or is object parsing part of this prompt (e.g., line 792-793)? If that's the case, that means the resultant object/part structures are actually **task-dependent**?


**Experiments**

It seems most experiments are just done in simulation environments. I think real-world experiments are important, ideally with some variations in object shapes/appearances/backgrounds, as the paper claims generalization. The objects involved in the sim experiments are mostly common daily-life objects, and with the setups in Section 5.3 (Appendix B), I think more quantitative evaluations on real robots won't take too much effort to conduct.

**Questions:**

See weaknesses. The questions mainly focus on an important point to be clarified: What are the classes of problems that the proposed framework can model and solve? Or in other words, the paper claims "generalization", then how "general" is the system?

---

### Official Review · Reviewer_FJ3j · 2025-10-31

**Soundness:** 2
**Presentation:** 2
**Contribution:** 1
**Rating:** 0
**Confidence:** 5

**Summary:**

This paper introduces GRACE, a framework designed to bridge the gap between the high-level semantic understanding of Vision-Language Models (VLMs) and the precise, low-level control required for robotic manipulation. The core idea is the introduction of "Executable Analytic Concepts" (EACs), a library of pre-defined, mathematically formulated blueprints that encode the geometry, affordances, and manipulation semantics of object parts.

**Strengths:**

The paper is well-written and clearly structured. The motivation for bridging the "semantic-to-physical" gap is well-articulated, and the proposed GRACE pipeline is explained in a logical sequence.

**Weaknesses:**

1. Over-engineered and Handcrafted Core: The entire framework is critically dependent on a pre-defined, manually curated library of "analytic concepts". The paper provides no discussion on the creation, maintenance, or scaling of this library. This approach does not solve the generalization problem but rather shifts it from learning a policy to the laborious, non-scalable task of engineering a comprehensive set of mathematical blueprints for every conceivable object part and interaction. This fundamentally limits the system's applicability in unstructured environments, as it can only handle objects that conform to its pre-existing library. Consequently, the claim of "zero-shot generalization" is narrow; the system generalizes to new instances of known concepts, but not to novel object categories or affordances for which a concept has not been manually defined.

2. Limited Novelty in Context of Existing Work: The idea of using structured representations to connect high-level reasoning with low-level control is a cornerstone of robotics research. The paper fails to sufficiently differentiate its approach from a large body of work that seeks to ground language in geometric and physical representations. For example, recent works use foundation models to reason about 3D scenes to generate motion plans or constraints directly, offering a more flexible paradigm than selecting from a rigid library (e.g., VoxPoser, ReKep from Huang et al. 2023, 2024). The proposed "analytic concepts" appear to be a hand-engineered analogue to learned representations like Neural Descriptor Fields (Simeonov et al., 2022), which also aim to capture object geometry and affordances in a structured, SE(3)-equivariant manner but do so by learning from data. The paper's dismissal of such data-driven approaches as suffering from "instability" is a strong claim that is not substantiated with direct comparative evidence.

3. Brittleness of the Multi-Stage Pipeline: The proposed system is a sequential cascade of multiple, distinct modules: task parsing, object segmentation, concept selection, parameter estimation, and motion planning. Such pipelines are notoriously brittle, as errors from any single stage can propagate and cause catastrophic failure downstream. The authors themselves concede that the primary sources of failure are pose estimation and inverse kinematics, which underscores the challenges of integrating disparate symbolic and geometric components without end-to-end optimization.

**Questions:**

1. Scalability of the Concept Library: Could the authors please provide more detail on the process and effort required to author a new analytic concept? For example, what would be involved in adding a concept for a "toggle switch" or a "spray bottle pump"? How many concepts were included in the library for the presented experiments, and how were they selected? A deeper understanding of this process is crucial for evaluating the practical scalability of the framework.

2. Justification Over Learning-Based Alternatives: The central thesis is that these analytic concepts are superior to learned representations. Could the authors provide a more direct or theoretical justification for this? For instance, why is this approach preferable to one where a VLM outputs parameters for a learned representation, such as a Neural Descriptor Field (Simeonov et al., 2022) or a keypoint-based model (e.g., kPAM from Manuelli et al., 2019), which might offer greater flexibility in representing non-canonical object shapes?

3. Handling Object Variation and Ambiguity: How does the system handle objects that do not neatly fit any of the pre-defined concepts in the library? For instance, how would it model an artisanal, irregularly shaped drawer handle? Does the parameter estimation module produce a high-error fit, or does the system have a mechanism to report a failure or lack of an appropriate concept?

4. Clarification of "Zero-Shot" Learning: The paper states that it performs "zero-shot generalization" but also mentions that the system includes "specialized MLP heads, each regressing a specific structural parameter" which are trained on a dataset generated from PartNet-Mobility. This suggests the system is trained for a known set of concepts. Could the authors please clarify their definition of "zero-shot"? Does it refer to unseen object instances and poses, or to entirely unseen object categories and manipulation skills for which no analytic concept exists?

---

### Official Review · Reviewer_GUzX · 2025-10-31

**Soundness:** 2
**Presentation:** 3
**Contribution:** 2
**Rating:** 2
**Confidence:** 3

**Summary:**

This paper proposes “Analytical Concepts”, an intermediate representation inferred by VLMs which can be leveraged to generate motion trajectories for manipulation without task demonstrations. It is mainly composed of three parts: geometric concept assets (for constructing geometric primitives), structural blueprint (for composing the geometric primitives to construct object instances using visual observations), and manipulation blueprint (for sampling grasp poses and post-grasp trajectories). The method is evaluated in both simulation (SimplerEnv and SAPIEN) and real world.

**Strengths:**

- It is an appealing high-level idea to construct object-centric concepts for manipulation and to use VLMs for inferring these concepts using visual observations
- The empirical results show that the proposed approach is competitive compared to baselines often trained on in-domain demonstration data.

**Weaknesses:**

- While it’s an appealing high-level idea, the specific method introduced in this work appears to be very specific to articulated objects. For example, the whole process of “geometric concept assets” and “structural blueprint”, albeit with their novel names, seems to be equivalent to the common procedure used for 3D reconstruction for articulated objects. For this purpose, many prior works have explored using VLMs in this context (such as https://xiahongchi.github.io/DRAWER/). Furthermore, this process appears to be not meaningful at all for rigid objects (or can be considered equivalent to 3D mesh reconstruction too). It’s unclear the specific novelty or benefits of the proposed approach (apart from it’s using VLMs which has been widely explored).
- The third module (Manipulation Blueprint) is also limited in its scope. It appears to always follow a fixed motion plan: collision-free motion to pregrasp pose → grasping → a post-grasp trajectory with push/pull wrench. While it’s perfectly valid if the scope is limited to only articulated objects, many practical manipulation tasks could involve other motions that are not being addressed, including non-prehensile manipulation, or more sophisicated post-grasp trajectory other than this point-vector parametrization.

**Questions:**

- In the provided demo, a long-horizon task is shown including placing multiple items into a tray. While the paper mentioned that long-horizon task is done by decomposing the instruction into discrete steps, it’s unclear why the robot intentionally places items early on to leave space for objects later using the proposed approach.

---

### Official Review · Reviewer_4MGW · 2025-11-02

**Soundness:** 3
**Presentation:** 3
**Contribution:** 2
**Rating:** 2
**Confidence:** 4

**Summary:**

This paper presents a new method that can generate physically executable actions in 3D environments.

The authors introduce Executable Analytic Concepts (EACs) — mathematically defined blueprints encoding: Object affordances (how something can be acted upon), Geometric constraints (shape, dimensions, layout), Manipulation semantics (e.g., “pull,” “twist,” “push”)

Then the authors introduce GRACE (Grounding Reasoning through Analytic Concept Execution), which integrates EACs into a structured policy pipeline. It shows great performance in both simulation environments and real world experiemtns.

There are some concerns and weakness of this framework:
1. It cannot handle challenging structure that cannot be handled by the EAC
2. All the policies are open-loop, which lack of failure recovery capability compared with VLA models.
3. Concept library coverage is finite and requires manual design
4. Performance depends on accurate pose and point-cloud estimation. This multi-stage policy might be bottlenecked by vision models.
5. Currently limited to two gripper primitives (push/pull), cannot handle cased like dexterous hand manipulation.

Also, there are many details that cannot be found in either main paper or supp materials, which makes the submission not sound enough.

**Strengths:**

1. The paper proposes a new concpet called EAC and the following pipeline, which bridges semantics and physics via mathematical blueprints.

2. EACs are transparent and reusable, which can be used for some specific cased and easy to debug.

3. It might be helpful for zero-shot generalization without task-specific training.

4. This pipeline is explainable, enabling human-readable and explainable robot behavior.

**Weaknesses:**

1. The framework struggles with complex or unconventional object structures that fall outside the representational capacity of the current Executable Analytic Concepts (EACs), reducing its effectiveness on highly irregular or deformable geometries.
2. All policies in GRACE operate in an open-loop manner, lacking feedback-based correction or failure recovery mechanisms. This limits robustness compared with closed-loop Vision-Language-Action (VLA) models that can adapt dynamically during execution.
3. The analytic concept library remains finite and relies on manual design, constraining scalability and adaptability to unseen object categories or novel task semantics.
4. The overall performance heavily depends on accurate pose and point-cloud estimation. Errors propagated from perception modules can significantly degrade downstream policy execution, making the multi-stage pipeline susceptible to bottlenecks in visual grounding.
5. The current implementation supports only two manipulation primitives—pushing and pulling—and is not yet capable of handling tasks requiring dexterous, multi-fingered hand control or fine-grained contact reasoning.

Also some details are missing:
1. please give the full object list and pictures of the real world experiments in the supp materials for sound experiment report.
2. please show the comparison of the baseline vla and your method, showcase why baseline method fails.
3. There is not details and pictures showing how the simulation experiments are performed. Please include more details.

**Questions:**

Please check weakness, provide the important details that are missing.

---

### Note · Authors · 2025-11-13

I have read and agree with the venue's withdrawal policy on behalf of myself and my co-authors.